# Blockade of Wnt Secretion Attenuates Myocardial Ischemia–Reperfusion Injury by Modulating the Inflammatory Response

**DOI:** 10.3390/ijms232012252

**Published:** 2022-10-14

**Authors:** Ingmar Sören Meyer, Xue Li, Carina Meyer, Oksana Voloshanenko, Susann Pohl, Michael Boutros, Hugo Albert Katus, Norbert Frey, Florian Leuschner

**Affiliations:** 1Internal Medicine III, University Hospital Heidelberg, 69120 Heidelberg, Germany; 2DZHK (German Centre for Cardiovascular Research), Partner Site Heidelberg-Mannheim, 69120 Heidelberg, Germany; 3German Cancer Research Center (DKFZ), 69120 Heidelberg, Germany

**Keywords:** myocardial infarction, monocytes, Wnt signaling, inflammation

## Abstract

Wnt (a portmanteau of *Wingless* and *Int-1*) signaling in the adult heart is largely quiescent. However, there is accumulating evidence that it gets reactivated during the healing process after myocardial infarction (MI). We here tested the therapeutic potential of the Wnt secretion inhibitor LGK-974 on MI healing. Ischemia/reperfusion (I/R) injury was induced in mice and Wnt signaling was inhibited by oral administration of the porcupine inhibitor LGK-974. The transcriptome was analyzed from infarcted tissue by using RNA sequencing analysis. The inflammatory response after I/R was evaluated by flow cytometry. Heart function was assessed by echocardiography and fibrosis by Masson’s trichrome staining. Transcriptome and gene set enrichment analysis revealed a modulation of the inflammatory response upon administration of the Wnt secretion inhibitor LGK-974 following I/R. In addition, LGK-974-treated animals showed an attenuated inflammatory response and improved heart function. In an in vitro model of hypoxic cardiomyocyte and monocyte/macrophage interaction, LGK974 inhibited the activation of Wnt signaling in monocytes/macrophages and reduced their pro-inflammatory phenotype. We here show that Wnt signaling affects inflammatory processes after MI. The Wnt secretion inhibitor LGK-974 appears to be a promising compound for future immunomodulatory approaches to improve cardiac remodeling after MI.

## 1. Introduction

Cardiovascular diseases (CVDs), including myocardial infarction (MI), are a leading cause of death around the globe. According to the World Health Organization (WHO) an estimated 17.9 million people die from CVDs each year, from which more than 50% of deaths result from MI [1]. In the last two decades, inpatient mortality following MI has declined tremendously due to improved treatment strategies and infrastructure. However, up to 36% of patients who survive acute MI will suffer from chronic heart failure (HF) [2,3], resulting in a significant socioeconomic burden [4]. 

MI is typically caused by plaque erosion or plaque rupture that results in the occlusion of a coronary artery. The occlusion leads to an insufficient blood supply to the underlying tissue and, consequently, to the death of cardiomyocytes. This initiates a complex healing process that replaces the lost tissue with a solid scar [5]. Clinical interventions aim to reperfuse the occluded artery to minimize the tissue damage due to the ischemic insult.

In recent years it became evident that an adequate inflammatory response is crucial for the healing process after MI. It is now well established that a prolonged and exaggerated inflammatory response can cause additional tissue damage, promote adverse remodeling, and is associated with poor prognosis [6]. 

Neutrophils are the first leukocyte subset that massively infiltrates the infarcted tissue within a few hours [7]. Shortly thereafter, monocytes and their lineage-descendent macrophages become the predominant immune cells [8,9]. Their response appears to be biphasic. The inflammatory Ly6Chi monocytes, with their proteolytic and phagocytotic properties, fulfill similar functions to neutrophils in the early infarct and clear cellular debris and damaged extracellular matrix. At later stages, reparative Ly6Clo macrophages promote angiogenesis and the deposition of extracellular matrix proteins to form a solid scar that replaces the lost heart tissue [8]. 

There is an increasing body of evidence that Wnt signaling may play an important role during the healing process after MI [10]. The name Wnt comes from the gene names *Wingless (Wg)*, an important gene during development in *drosophila*, and its murine homolog *Int-1*, which is a known protooncogene. Although largely quiescent in the healthy adult heart, Wnt signaling gets reactivated upon myocardial injury [11]. Wnt signaling can be roughly subdivided into a *β-catenin*-dependent canonical and a *β-catenin*-independent non-canonical signaling pathway. The latter uses *c-Jun-terminal kinases (JNK)* and the transcription factor *activating transcription factor 2 (ATF2)*, among others, for signal transduction [12]. We could previously show that infiltrating monocytes are activated during myocardial infarction via the non-canonical Wnt signaling pathway at the site of injury. Furthermore, inhibition of Wnt signaling by the extracellular Wnt inhibitor *Wnt inhibitory factor-1 (WIF-1)* leads to improved infarct healing by modulating the inflammatory response [13].

Secretion of Wnt proteins requires post-translational modifications by the endoplasmic reticulum transmembrane O-acyltransferase *Porcupine (Porcn)*. Inhibition of *Porcn* by the small molecule LGK-974 has been shown to effectively inhibit Wnt protein secretion [14]. The inhibitor is currently in clinical trials for cancerous diseases that show dysregulated Wnt signaling.

We here evaluated whether inhibition of Wnt protein secretion by the *Porcn* inhibitor LGK-974 modulates the inflammatory response and improves the healing process in a murine model of MI (ischemia–reperfusion (I/R) model). We found that treatment with LGK-974 attenuates the monocytic response and improves the healing process following MI.

## 2. Results

### 2.1. RNAseq Analysis Reveals Altered Gene Expression of Inflammation-Associated Genes upon LGK974 Administration

In order to elucidate the biological processes that are influenced by Wnt signaling during myocardial infarction, we treated animals with the Wnt secretion inhibitor LGK-974 and compared the transcriptomes with vehicle-treated (dimethyl sulfoxide (DMSO)) animals (a timeline of the experimental procedure and the animal use is depicted in Figure 1). 

RNA-seq analyses revealed that LGK-974 treatment results in a diverse transcriptional response in the infarcted area. We observed differential expression of 529 genes in infarcted heart tissue isolated from either LGK-974 or DMSO-treated animals (Figure 2). 

Gene set enrichment analysis (GSEA) revealed that a great proportion of differentially expressed genes are associated with the immune system and its function or profibrotic processes (Table 1).

### 2.2. Blockade of Wnt Protein Secretion by LGK-974 Attenuates the Inflammatory Response after MI

Prolonged and exaggerated inflammatory responses after cardiac injury can cause additional tissue damage and promote adverse remodeling [6]. The Canakinumab Anti-Inflammatory Thrombosis Outcome Study (CANTOS) demonstrated that patients with coronary heart disease benefit from immunomodulating therapy [15]. We, therefore, evaluated the effect of LGK-974 on the inflammatory response following MI. 

We next evaluated the inflammatory response two days after induction of I/R injury using flow cytometry. Figure 3a represents the gating strategy for cell debris and doublet exclusion. We observed reduced infiltration of leukocytes in the infarcted hearts of LGK-974 treated animals compared to the vehicle control (Figure 3b,d). Since the myeloid cell response plays a major role in infarct healing, we further evaluated cell numbers of monocyte/macrophage-subsets and neutrophils in infarcted hearts. The myeloid response was generally reduced in LGK-974 treated animals compared to the controls (Figure 3c,e). Further, flow cytometric analysis revealed that inflammatory monocytes numbers were significantly reduced (Figure 3f–h), while a clear trend was observed for macrophage and neutrophil numbers between LGK-974 and DMSO-treated animals, no statistically significant difference was observed (Figure 3i,j). We further determined cardiac *troponin T* levels 24 h after induction of I/R injury to test if LGK-974 already has an effect on initial infarct sizes. However, we did not observe differences between LGK-974 and DMSO-treated animals (Figure 3k).

### 2.3. Blockade of Wnt Secretion by LGK-974 Preserves Heart Function 

Inflammatory reactions can contribute to adverse remodeling and may contribute to the onset of heart failure after MI [16]. Therefore, we evaluated if the anti-inflammatory effect observed due to LGK-974 treatment also impacts heart function 28 days after induction of I/R injury. For the echocardiographic evaluation of the heart function and the histological evaluation, 4 weeks after the induction of I/R-injury we operated 28 animals in total. In the DMSO control group, 2 animals died, whereas in the LGK-974 treated group, 3 animals died. However, the animals died immediately after or during the surgery; therefore, we concluded that the mortality was independent of the treatment. An additional 4 animals had no elevation in *troponin T* levels, i.e., no ischemia–reperfusion injury, and were excluded from the analysis. LGK-974 treatment preserves heart function following I/R injury (see Figure 4a for representative M-Mode images of DMSO and LGK-974 treated animals). Animals, who received a daily dose of LGK-974 during the inflammatory phase (day 1 to day 7, post-injury) had a higher fractional shortening and a higher ejection fraction compared to vehicle-treated animals (Figure 4b,c). LGK974 treatment had no effect on heart function under basal conditions (see Appendix A). The heart rate between the LGK-974 and DMSO-treated animals did not differ 28 days after the induction of I/R injury (see Appendix A). In addition, LGK974 treatment resulted in reduced fibrosis in the LV wall (Figure 4d,e). Taken together, modulation of Wnt secretion during the inflammatory response after MI may be beneficial and could improve cardiac remodeling.

### 2.4. Inhibition of Wnt Secretion Inhibits Monocyte/Macrophage Activation In Vitro

We next tried to elucidate how LGK-974 may alter the inflammatory process in vitro. For this purpose, we made use of an in vitro model for studying cardiomyocyte-monocyte/macrophage interactions. We cultured HL-1 heart muscle cells under hypoxic conditions (1.5% O_2_) in the presence or absence of LGK-974 for 24 h. Western blot analysis of the supernatant revealed that LGK-974 effectively reduced the secretion of Wnt5a by cardiomyocytes (see Appendix A). We could also show that the supernatant of hypoxic cardiomyocytes triggers ATF2 phosphorylation (see Appendix A). We then stimulated monocytes/macrophages with the supernatant for 4 h. Monocytes/macrophages that were stimulated with the supernatant of cardiomyocytes that were cultured under hypoxic conditions in the presence of LGK-974 showed reduced activation of non-canonical Wnt signaling, as seen by reduced ATF2 phosphorylation (Figure 5a,b). In addition, these monocytes/macrophages showed reduced expression levels of inflammatory markers such as TNFα, IL-1β (Figure 5c), and MMP2 measured by qPCR (Figure 5d top); MMP9, however, remained unaltered (Figure 5d bottom), indicating that secreted Wnt proteins may be used by cardiomyocytes to stimulate an immune response.

## 3. Discussion

Myocardial infarction elicits a strong stimulation of the immune system leading to massive infiltration of myeloid cells into the infarcted heart [17,18]. We and others could previously show that immune cells that infiltrate the infarcted heart get activated via the non-canonical Wnt signaling pathway. Inhibition of Wnt signaling by extracellular inhibitors had anti-inflammatory effects and improved infarct healing [13,19]. We here inhibited Wnt protein secretion with the small molecule LGK-974 during the inflammatory response after MI. Our finding further supports the hypothesis that adjustment of Wnt signaling attenuates inflammatory (especially the monocytic) responses and improves the healing process. 

Our RNA-seq analysis and GSEA show that the inhibition of Wnt protein secretion leads to alterations in the expression of genes that are associated with the immune system. However, since RNAseq was performed on tissue (from the infarcted area), we cannot pin down these findings to specific cell types or processes. The finding that other gene sets play a role in collagen formation and extracellular matrix organization suggests that Wnt signaling interferes with many cellular processes, such as fibrosis, and supports previous findings [20]. Our findings further support the anti-fibrotic effects of LGK-974 recently described by Moon et al. in a model of permanent ligation of the LAD [21]. We here investigated temporary coronary occlusion, which might reflect the clinical situation a bit better.

Our findings indicate that treatment with LGK-974 during the inflammatory phase attenuates the infiltration of immune cells, especially of monocytes, to the site of injury. It has been reported, however, that inhibition of Wnt signaling in stressed cardiomyocytes can reduce apoptosis [22]. Furthermore, overexpression of the canonical Wnt mediator β-catenin results in increased inflammatory cytokine production in cardiomyocytes [23]. A reduced inflammatory response due to inhibition of Wnt signaling could be a secondary effect due to potentially reduced loss of cardiomyocytes or their secretion of pro-inflammatory proteins. However, we did not measure statistically significant differences in cardiac *troponin T* plasma levels, which are directly correlated to the damage of heart muscle cells. 

It has been previously reported that Wnt signaling is activated in the bone marrow following MI leading to enhanced mobilization and increased numbers of hematopoietic progenitor cells [24]. Since LGK-974 was administered systemically, the inhibition of Wnt signaling in the bone marrow niche could also have led to decreased numbers of immune cells. 

Others have also reported that inhibition of Wnt signaling reduces the infiltration of neutrophils to the infarcted heart [25]. However, we did not observe significantly altered neutrophil numbers in the infarcted hearts. Neutrophils peak early after the ischemic insult. We were analyzing the inflammatory response two days after the I/R injury when neutrophil numbers could already be on the decline. The effect size of LGK974 treatment on neutrophil numbers could also be lower than observed for monocyte/macrophage numbers. A significant reduction of neutrophil numbers, therefore, might have been observed with an increased sample size. 

We further evaluated how LGK-974 administration leads to altered inflammatory responses. Our in vitro experiments support the hypothesis that Wnt signaling is used in cell–cell communication to activate immune cells. The supernatant from hypoxic cardiomyocytes that were treated with LGK-974 activates monocytes/macrophages to a lesser extent compared to the supernatant from vehicle-treated cardiomyocytes. In our set up, we cannot exclude that residual LGK974 in the supernatant has a direct effect on monocytes/macrophages. Information on Wnt protein production in specific cell types is scarce [26]. However, Palevski et al., for instance, have shown that loss of Wnt secretion by macrophages also has beneficial effects on infarct healing [27]. In addition, Wnt secretion has been shown to promote myofibroblast formation and the progression of fibrosis during experimental autoimmune myocarditis [28]. Wnt secretion may therefore be important in a unidirectional way and a variety of cell-cell communication processes and may also be used in an autocrine way.

We here focused on modulation of the non-canonical Wnt signaling pathway in monocytes/macrophages since we previously found that monocyte/macrophages get activated via the non-canonical Wnt signaling pathway in the infarcted heart. However, the porcupine inhibitor LGK-974 will most likely also inhibit the secretion of Wnt proteins associated with the canonical Wnt signaling pathway. Inhibition of the canonical Wnt signaling pathway was also shown to have immunomodulating properties (nicely reviewed by [29]. Activation of the canonical Wnt signaling pathway by inhibition of GSK3β, for instance, has been shown to promote the differentiation into anti-inflammatory macrophages [30]. Administration of LGK-974 with regards to canonical Wnt signaling inhibition could therefore have negative effects on the resolution of the inflammatory phase. Determining the time course of Wnt signaling activation of the distinct branches in immune cells could help to improve treatment strategies. 

We assume that other orally administered Porcn inhibitors, like ETC-1922159 and CGX1321, may have similar effects to LGK-974, which has been described in clinical trials for cancer treatments [31]. It has yet to be determined which Wnt receptor complexes and Wnt signaling pathways, with their distinct branches, are the driving force in monocyte/macrophage activation. However, blocking Wnt receptors with monoclonal antibodies such as vantictumab and ipafricept (a fusion protein consisting of the extracellular ligand-binding domain of FZD8 and the IgG1 Fc fragment) may also be an interesting treatment strategy in myocardial infarction. Furthermore, we and others have demonstrated that a blockade of Wnt ligands with extracellular Wnt inhibitors like WIF1 and SFRP5 also improves the healing process of myocardial infarction by modulating the inflammatory response [13,19]. A better understanding of Wnt signaling activation is required to develop a “tailored” treatment strategy that specifically targets the relevant components. 

We could show that LGK974 reduces the secretion of the non-canonical Wnt ligand Wnt5a from cardiomyocytes, which could contribute to the anti-inflammatory effects. However, we cannot exclude that inhibiting the secretion of other non-canonical Wnt signaling-associated ligands may also be involved. The non-canonical Wnt9A and B, for instance, have been shown to be upregulated in a cardiac cryoinjury model in neonatal mice [32]. Wnt9A has been shown to correlate with pro-inflammatory cytokines [33]. The full Wnt ligand secretion pattern in cardiomyocytes still needs to be elucidated.

Finally, we evaluated the effect of LGK-974 on heart function 28 days after the induction of I/R injury when the remodeling process after the acute ischemic event is completed. Many patients who survive acute MI suffer from heart failure subsequently. We found that heart function was greatly improved when LGK-974 was administered during the inflammatory response after MI underscoring the therapeutic potential of LGK-974 for the treatment of MI. 

In summary, this study evaluated the potential of LGK-974 as an immunomodulating drug to improve the healing process after myocardial infarction. We demonstrated that LGK-974 preserves heart function and may improve cardiac healing following myocardial infarction. We further demonstrated that LGK-974 might have immunomodulating properties by inhibiting Wnt signaling in immune cells. Activation of Wnt signaling in other inflammatory conditions, such as sepsis, has also been reported to be detrimental [34]. Inhibition of Wnt signaling, in turn, appears to be beneficial to the outcome [35]. The anti-inflammatory and Wnt signaling modulating properties of LGK-974 could, therefore, be translated to other pathological conditions, such as sepsis or cancer [36]. 

## 4. Materials and Methods

### 4.1. Induction of Ischemia–Reperfusion Injury and LGK-974 Treatment

Ischemia/reperfusion (I/R) injury was induced in female C57BL/6 mice (Janvier, Saint-Berthevin, France) aged 10–12 weeks. In brief, anesthesia was induced with isoflurane (4%, 600 mL O_2_/min) and maintained by endotracheal ventilation (1.5–4%, 600 mL O_2_/min). Thoracotomy was performed in the fourth left intercostal space. The left ventricle was exposed, and the left anterior descending (LAD) artery was occluded for 45 min using suture material. After suture removal, chest and skin were closed and anesthesia was terminated. Mice were extubated when breathing was restored. Animals received directly after extubation an oral gavage of 3 mg/kg LGK-974 (Selleck Chemicals, Houston, TX, USA) followed by a daily dose of 3 mg/kg until day 7, following I/R surgery. DMSO was used as a solvent for LGK-974 (LGK-974 stock solution 5 mg/mL). LGK-974 was formulated in 10% (*v*/*v*) citrate buffer (pH 2.8)/90% (*v*/*v*) citrate buffer (pH 3.0). Control animals received the DMSO vehicle in citrate buffer solution only. See also Liu et al., PNAS 2013). An oral gavage dose had a total volume of up to 100 µL. LGK-974 was given until day 7 after induction of I/R Injury to target the inflammatory/immune response. 

### 4.2. RNA-Sequencing Analyses

For RNA-sequencing analyses tissue from the infarct area was harvested 2 days after induction of ischemia–reperfusion injury and total RNA was isolated using Trizol reagent (Thermo Fischer Scientific, Waltham, MA, USA), according to the manufacturer’s instructions. RNA concentration was quantified using Qubit fluorometer (Invitrogen, Carlsbad, CA, USA). RNA quality was further assessed using BioAnalyzer (Agilent Technologies, Santa Clara, CA, USA). Ribosomal RNA was depleted using RiboZero (Illumina, San Diego, CA, USA). RNA-seq libraries were generated using TrueSeq RNA Access Library Prep Kit (Illumina, San Diego, CA, USA). Sequencing was performed on HiSeq2000 sequencer (Illumina, San Diego, CA, USA).

Abundances of transcripts were quantified by using Kallisto [37]. A transcriptome index for pseudoalignment was built with the ensemble genome assembly GRCm38. Differential gene expression analysis was carried out with the DESeq2 package [38]. Gene set enrichment analysis was performed using the reactome pathway database [39].

### 4.3. Flow Cytometry Analysis

Single-cell suspensions from whole heart tissue were obtained by mincing the tissue with fine scissors. Minced tissue was then further digested with a solution containing collagenase I, collagenase XI, hyaluronidase (Sigma-Aldrich Chem GmbH, Taufkirchen, Germany), and DNase I (BD Biosciences, Heidelberg, Germany). Solution was then passed through a 70 µm cell strainer (BD Biosciences, Heidelberg, Germany). 10 × 10^6^ cells per sample were stained. Leukocytes were identified as Lin−(CD90; B220;CD49b; NK1.1; Ly6G; Ter119); CD45+. Myeloid cells were identified as Lin−(CD90; B220; CD49b; NK1.1; Ly6G; Ter119); CD45+; CD11b+. Inflammatory monocytes were identified as Lin−(CD90; B220; CD49b; NK1.1; Ly6G; Ter119); F4/80−; CD11c−; CD11b+; Ly6Chi. Macrophages were identified as Lin−(CD90; B220; CD49b; NK1.1; Ly6G; Ter119); F4/80+; CD11c−; CD11b+; Ly6Clo. as Neutrophils were identified as Lin+; CD11b+; F4/80−; CD11c−; Ly6Cint.

### 4.4. Histology

Hearts were extracted 4 weeks after induction of I/R injury. Hearts were rinsed in PBS and fixed for 1 h in 10% formaline. Hearts were then cut in half along the long axis and further fixed for another 23 h and embedded in paraffin. Serial 5 µm sections were stained with Masson’s trichrome staining (Sigma-Aldrich Chem GmbH, Taufkirchen, Germany) according to the manufacturer’s instructions. The percentage of the fibrotic area of the left ventricular (LV) wall was determined using ImageJ. 

### 4.5. Ethical Statement

Animal studies were approved by the regulatory authorities (Regierungspräsidium Karlsruhe of the state of Baden-Württemberg/Germany, 35-9185.81/G-167/16). Animals were euthanized by cervival dislocation. A total of 52 were used in this study.

### 4.6. High-Sensitive Troponin T Determination

Plasma samples were collected 24h after the induction of I/R injury. To evaluate initial infarct sizes *troponin T* levels were measured by electrochemiluminescence (Elecsys 2010 analyzer, Roche Diagnostics, Germany). Plasma samples were diluted (1:20) with PBS.

### 4.7. In Vitro Experiments

To validate our in vivo findings and to further evaluate the immunomodulating properties of LGK-974 on monocytes/macrophages, we performed further experiments in our in vitro model of hypoxic cardiomyocyte and monocyte/macrophage interaction. HL-1 cardiomyocytes were passaged and subcultured prior to treatment as previously described [40]. For in vitro experiments, HL-1 cardiomocytes were cultured under hypoxic conditions (1.5% O_2_) in serum-free Claycomb medium (Sigma-Aldrich Chem GmbH, Taufkirchen, Germany) supplemented with 2 mM L-Glutamine and 1% Penicillin/Streptomycin (Thermo Fisher Scientific, Waltham, MA, USA) in the presence or absence of 10 µM LGK974 for 24 h. LGK974 concentrations were chosen at the lower range of the concentrations suggested by the manufacturer and described in the original publication [14]. The supernatant was recovered. 

For analysis of Wnt5a secretion in cardiomyocyte supernatant, blue Sepharose beads (Sigma Aldrich Chem GmbH, Taufkirchen, Germany) were added to the supernatant and rotated overnight at 4 °C. The beads were then spun down and resuspended in standard 1x laemmli buffer/1x RIPA buffer and boiled for 5 min at 95 °C. The beads were spun down again, and the supernatant was loaded on SDS-PAGE for Western blot analysis. 

RAW 264.7 macrophages (ATCC, Manassas, VA, USA) were cultured prior to the experiments according to the manufacturer’s instructions. In in vitro experiments, macrophages were stimulated with the supernatant of hypoxic HL-1 cardiomyocytes for 4 h. Proteins were isolated with RIPA buffer (Cell signaling technology, Danvers, MA, USA) according to manufacturer’s instructions. 

### 4.8. Echocardiography

Echocardiography was performed on a Vevo 2100 (VisualSonics, Toronto, ON, Canada) 4 weeks after induction of I/R injury. Mice were conscious during all echocardiographic measurements. Ejection fraction (EF) and fractional shortening (FS) were determined based on M-mode measurements.

### 4.9. Quantitative Real-Time PCR

Total RNA was isolated from cells using Trizol reagent (Thermo Fischer Scientific, Waltham, MA, USA) according to the manufacturer’s instructions. 1 µg RNA was reverse transcriped using Revert Aid First Strand cDNA Synthesis kit (Thermo Fisher Scientific, Waltham, MA, USA) according to manufacturer’s instructions. Gene expression was analyzed using quantitative real-time PCR (qPCR) using SYBR Green (Bio-Rad Laboratories, München, Germany) according to the manufacturer’s instructions on an Applied Biosystems 750 real-time PCR system. The following primers were used: TNFα: tcttctcattcctgcttgtgg (fwd); ggtctgggccatagaactga (rev) MMP2: acgagcgaagggcatacaaa (fwd); accctagggatgcttggat (rev). IL-1β: tcgtgctgtctgacccatgt (fwd), acaaagctcatggagaataccact (rev); MMP9: ctctcctggctttcggctg (fwd), agcggtacaagtatgcctctgc (rev). Expression was normalized to HPRT: gtcaacgggggacataaaag (fwd); tgcattgttttaccagtgtcaa (rev). 

### 4.10. Western Blotting

Protein lysates of cells were prepared using RIPA buffer supplemented with a proteinase inhibitor cocktail (Cell Signaling Technology, Danvers, MA, USA). Protein concentrations were measured using a BCA assay (Thermo Fisher Scientific, Waltham, MA, USA). Equal amounts of protein were separated on 4–15% SDS-PAGE gradient gels (Bio-Rad Laboratories, München, Germany) and transferred to PVDF membranes (Merck Chemicals, Darmstadt, Germany). The following first antibodies monoclonal rabbit anti-GAPDH [EPR16891] (Abcam, Cambridge, UK); anti-ATF2 (phosphor T51 + T69) (Abcam, Cambridge, UK), anti-total ATF2 (Abcam, Cambridge, UK); anti-Wnt5A (Abcam, Cambridge, UK) and HSC70 (Santa Cruz, Dallas, TX, USA) were used. Protein expression was analyzed using Image J. 

### 4.11. Statistics

Statistical analyses were performed using GraphPad Prism 7 (GraphPad Software Inc., La Jolla, CA, USA). Data are represented as mean ± SD. Differences between the two groups were analyzed by Student’s *t*-test with correction for multiple testing using Dunn–Sidak method. *p* < 0.05 was considered to be statistically significant.

## Figures and Tables

**Figure 1 ijms-23-12252-f001:**
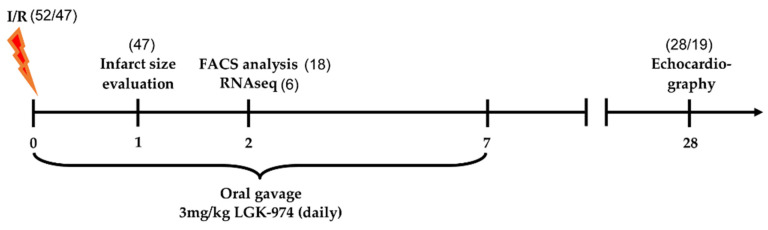
Timeline and animal use of in vivo experiments. Numbers in brackets represent the animals used in the experiments. In total, 52 mice underwent ischemia/reperfusion (I/R) surgery, of which five animals died during or shortly after the procedure. Blood was drawn from the surviving animals (47) for *troponin T* measurements. 18 (9 per group) animals were sacrificed for flow cytometric analysis, 6 (3 per group) were sacrificed for RNA seq experiments. In total, 19 animals (8 in the DMSO group and 11 in the LGK-974 group) were used for echocardiographic and histological evaluation; four animals had no elevated *troponin T* levels and were excluded from the analysis.

**Figure 2 ijms-23-12252-f002:**
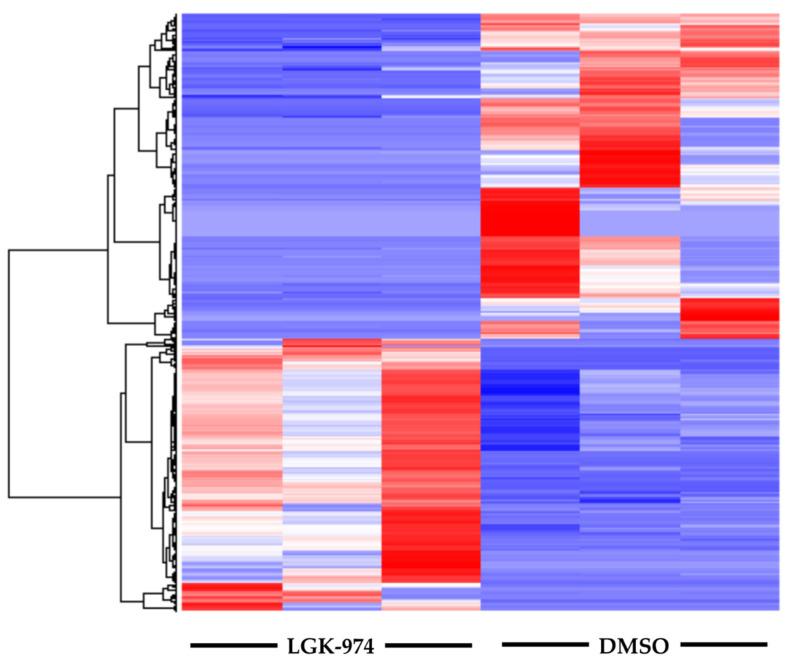
Heatmap of differentially expressed genes. Columns represent individual samples from the two groups. RNA-seq analysis revealed differential expression of 529 genes in heart tissue (from the infarct area) from animals treated with LGK-974 compared to DMSO (vehicle control). N = 3 per group.

**Figure 3 ijms-23-12252-f003:**
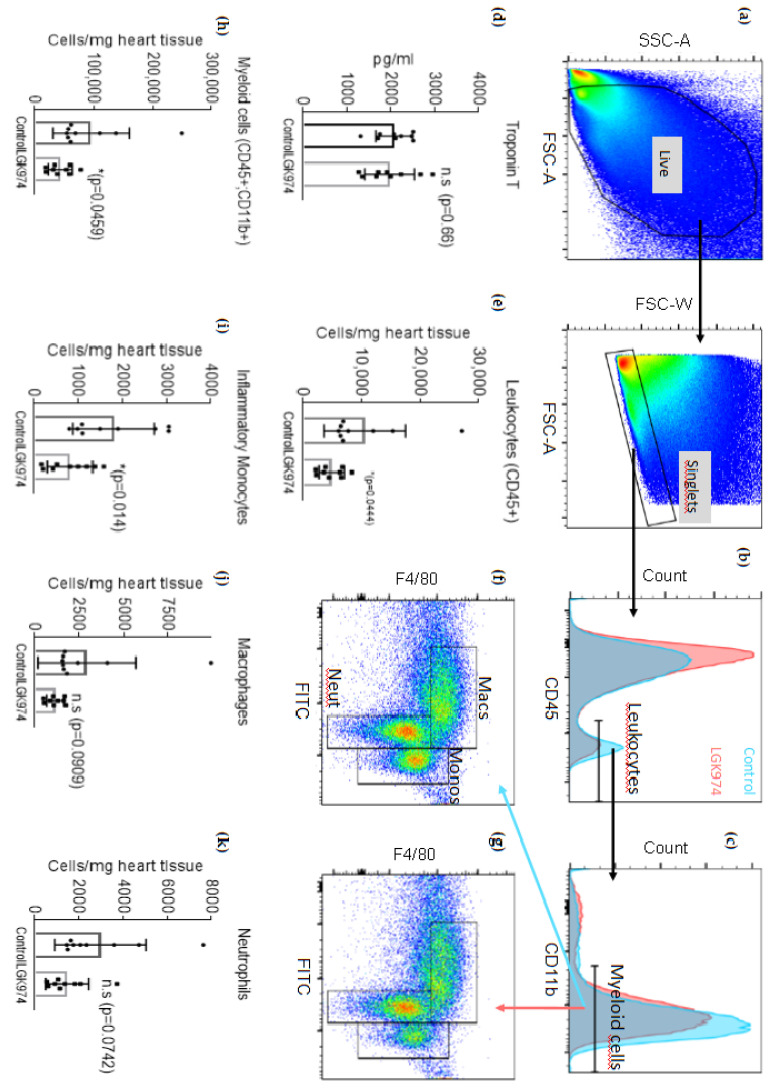
LGK-974 treatment alters the inflammatory response following I/R injury. (**a**) Gating scheme excluding dead cells, cell debris, and doublets. (**b**,**c**) representative histogram of heart tissue leukocytes resp. myeloid cells 2 days after I/R injury. (**d**) Quantification of total leukocytes in heart tissue following I/R injury. (**e**) Quantification of myeloid cells in the heart following I/R injury. (**f**) and (**g**) representative images of flow cytometric analysis of heart tissue cell suspension following I/R injury in DMSO resp. LGK-974 treated animals on monocytes, macrophages, and neutrophils. (**h**–**j**) quantification of inflammatory monocytes, macrophages, and neutrophils two days after I/R injury. (**k**) cardiac *troponin T* plasma levels 24 h after I/R injury. N = 9 per group.

**Figure 4 ijms-23-12252-f004:**
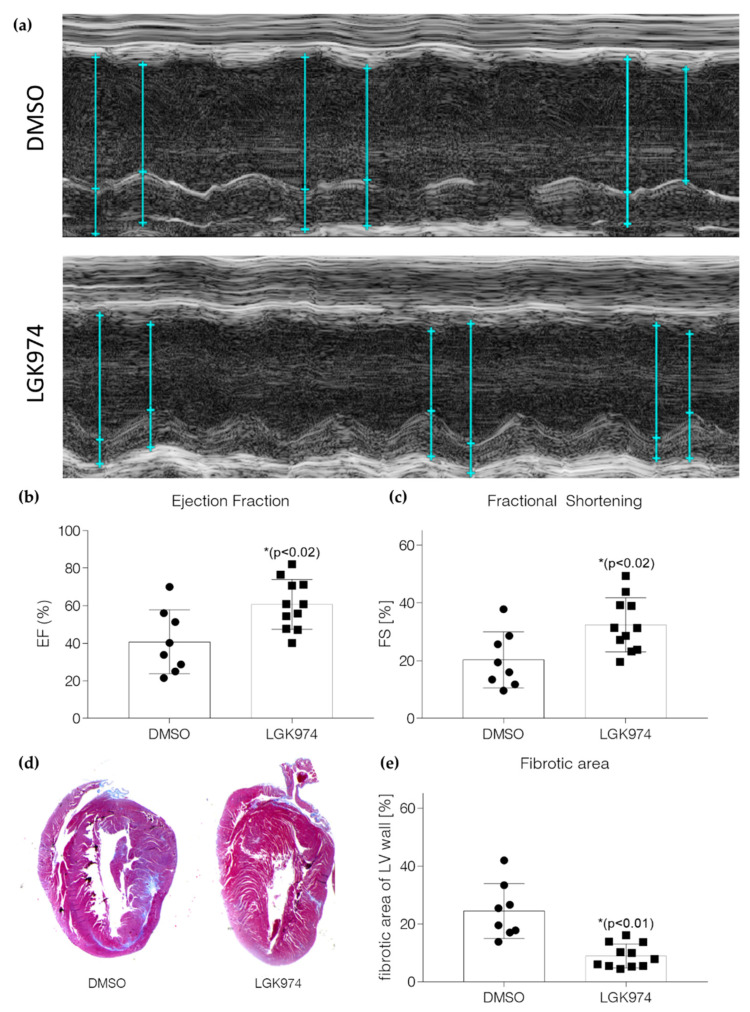
LGK-974 treatment during the inflammatory response following I/R injury improves heart function. (**a**) Representative M-Mode echocardiographic images of DMSO (top) and LGK-974 treated (bottom) animals. (**b**,**c**) Echocardiographic results from DMSO and LGK-974 treated animals. (**d**) Representative Masson trichrome staining of DMSO (left) and LGK974 (right) treated animals 4 weeks after I/R injury. (**e**) Quantification of the fibrotic area of the LV wall in DMSO resp. LGK974 treated animals. N = 8 for DMSO-treated animals and N = 11 for the LGK-974-treated animals.

**Figure 5 ijms-23-12252-f005:**
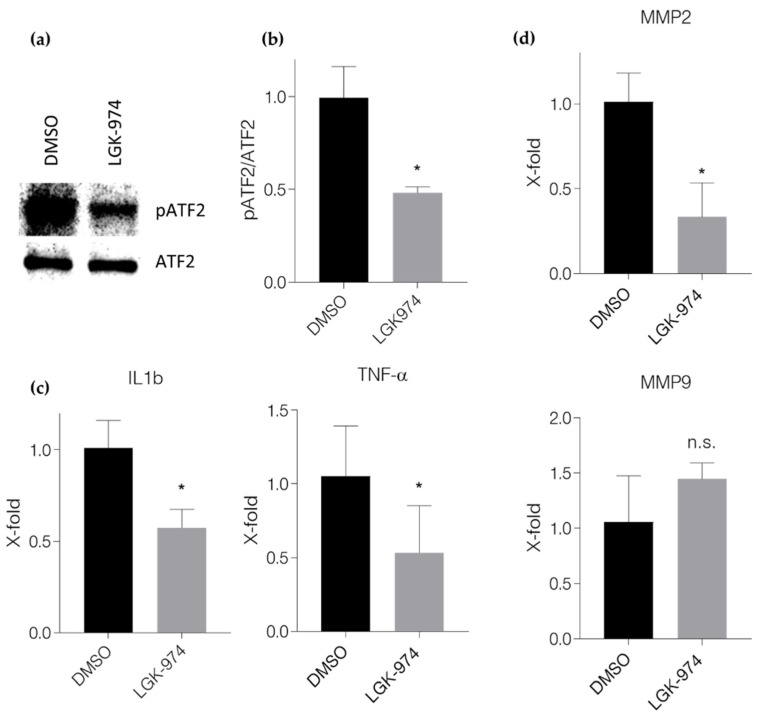
LGK-974 inhibits non-canonical Wnt signaling in monocytes/macrophages and leads to a reduced inflammatory character. (**a**) representative Western blots of pATF2 expression in monocytes/macrophages stimulated with supernatant of heart muscle cells that were cultured under hypoxic conditions in the presents or absence of LGK-974 and (**b**) quantification. (**c**) Expression of the inflammatory markers TNFα and IL1β and (**d**) the matrix metalloproteinases MMP2 (top) and MMP9 (bottom) measured by qPCR in monocytes/macrophages stimulated with supernatant of heart muscle cells that were cultured under hypoxic conditions in the presents or absence of LGK-974. Results originate from three independent experiments. n.s.: not significant; *: *p* ≤ 0.05.

**Table 1 ijms-23-12252-t001:** Top 10 hits of GSEA. We found 529 differential expressed genes in heart tissue from LGK-974 compared to DMSO-treated animals. Using GSEA we identified classes of genes that are over-represented in our differential expressed gene set. Many genes that were differentially expressed were either associated with immunity or participated in fibrotic processes (e.g., extracellular matrix (ECM) organization and collagen formation). Genesets are derived from the reactome pathways database.

Gene Set Name	Description	Genes	FDR
IMMUNE_SYSTEM	Genes involved in immune system	25	6.41 × 10^−6^
ADAPTIVE_IMMUNE_SYSTEM	Genes involved in adaptive immune system	19	3.02 × 10^−5^
DEVELOPMENTAL_BIOLOGY	Genes involved in developmental biology	18	2.84 × 10^−6^
AXON_GUIDANCE	Genes involved in axon guidance	15	2.68 × 10^−6^
EXTRACELLULAR_MATRIX_ORGANIZATION	Genes involved in extracellular matrix organization	6	6.8 × 10^−3^
PLATELET_HOMEOSTASIS	Genes involved in platelet homeostasis	6	4.74 × 10^−3^
COLLAGEN_FORMATION	Genes involved in collagen formation	6	1.00 × 10^−3^
SIGNALING_BY_ROBO_RECEPTOR	Genes involved in signaling by Robo receptor	5	6.26 × 10^−4^
TCR_SIGNALING	Genes involved in TCR signaling	5	6.8 × 10^−3^
SEMAPHORIN_INTERACTIONS	Genes involved in semaphorin interactions	5	1.77 × 10^−2^

## Data Availability

RNA sequencing data will be available in the GEO database repository (GEO submission GSE185444).

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
