# Peer review of "Blockade of Wnt Secretion Attenuates Myocardial Ischemia–Reperfusion Injury by Modulating the Inflammatory Response"

_ijms, 2022, doi:10.3390/ijms232012252_

Round 1

Reviewer 1 Report

This is a research paper demonstrating that inhibition of wnt signaling reduces IR injury in the mouse model, reducing inflammation and preserving LV function. The study is well prepared, executed and reported and the conclusions are based on obtained results.  I have only some minor comments:

1. “Animals received directly after surgery an oral gavage of 3mg/kg LGK-974” is unclear. Does the “surgery” refer to induction of ischemia, reperfusion or closing of the chest?

2. Hypoxic culture of cardiomyocytes is not a model of myocardial infarction; it can merely be regarded as a model of hypoxic injury. Myocardial infarction is much more than hypoxia.

3. How was the infarct area for RNA sequencing analysis identified?

Author Response

Reviewer 1

We thank the reviewer for carefully evaluating our manuscript and the valuable comments. We like to address the comments as followed:

  1. Animals received directly after surgery an oral gavage of 3mg/kg LGK-974” is unclear. Does the “surgery” refer to induction of ischemia, reperfusion or closing of the chest?

We apologize for the inaccuracy in describing the methods. We temporarily occluded the LAD, closed the chest and ended the isoflurane anesthesia. When the animals started to breath by their own, we extubated the animals and administered an oral gave of LGK-974. We changed the description as followed:

“Ischemia/reperfusion (I/R) injury was induced in female C57BL/6 mice (Janvier, Saint-Berthevin, France) aged 10-12 weeks. In brief, anesthesia was induced with isoflurane (4%, 600ml O2/min) and maintained by endotracheal ventilation (1,5% - 4%, 600ml O2/min). Thoracotomy was performed in the fourth left intercostal space. The left ventricle was exposed and the left anterior descending (LAD) artery was occluded for 45min using suture material. After suture removal, chest and skin were closed and anesthesia was terminated. Mice were extubated when breathing was restored. Animals received an oral gavage of 3mg/kg LGK-974 (Selleck Chemicals, Houston, USA) directly after extubation, followed by a daily dose of 3mg/kg until day 7 following I/R surgery.”

  1. Hypoxic culture of cardiomyocytes is not a model of myocardial infarction; it can merely be regarded as a model of hypoxic injury. Myocardial infarction is much more than hypoxia.

We fully agree that an in vitro model cannot mirror the whole spectrum of myocardial infarction. We here aimed to mimic the interaction of hypoxic cardiomyocytes with monocytes/macrophages. Of course, these are only two of many cell types involved and only one aspect of this multifaceted pathology. The name was chosen for simplification. In order to correct this misleading term, we renamed the in vitro model to: “in vitro model of hypoxic cardiomyocyte and monocyte/macrophage interaction”.

  1. How was the infarct area for RNA sequencing analysis identified?

The infarcted area for RNA sequencing was identified by our highly experienced surgeon/scientist who has an experience of inducing LAD ligation for > 5years. After tissue harvest the punctures from the temporal occlusion of the LAD are still visible. The tissue for RNA sequencing was extracted below the punctures from the occlusion. However, a certain distance was kept to exclude tissue that was damaged due to the surgery. Furthermore, the chosen tissue was limited in width to exclude viable/border zone tissue.

Reviewer 2 Report

1. The authors may need to discuss whether other Wnt signaling inhibitors can achieve the same results as what LGK974 did. 

2. In discussion, since there are 19 members in Wnt family, how many potential non-canonical Wnt ligands may be possibly involved in the inhibitory effects of LGK974 anti-inflammatory response?

3. Canonical Wnt signaling pathway may need to be detected in the in vitro cell model, such as the change of beta-catenin level before and after LGK974 treatment. 

Author Response

Reviewer 2

We thank the reviewer for the valuable suggestions to improve our manuscript.

  1. The authors may need to discuss whether other Wnt signaling inhibitors can achieve the same results as what LGK974 did. 

This is indeed an interesting point to discuss. There are various pharmacological Wnt signaling inhibitors (and Wnt secretion inhibitors) available. Some of them have a similar mode of action. Others target  However, we can only speculate on this matter due to the complexity of Wnt signaling. We added the following paragraph regarding other Wnt signaling inhibitors to the discussion:

We assume that other orally administered Porcn inhibitors, like ETC-1922159 and CGX1321, may have similar effects like LGK-974, which has been described in clinical tri-als for cancer treatments [31]. It has yet to be determined, which Wnt receptor complexes and Wnt signaling pathways with their distinct branches are the driving force in mono-cyte/macrophage activation. However, blocking Wnt receptors with monoclonal antibod-ies such as Vantictumab and ipafricept (a fusion protein consisting of the extracellular ligand-binding domain of FZD8 and the IgG1 Fc fragment) may also be an interesting treatment strategy in myocardial infarction. Furthermore, we and others have demon-strated that a blockade of Wnt ligands with extracellular Wnt inhibitors like WIF1 and SFRP5 also improve the healing process of myocardial infarction by modulating the in-flammatory response [13, 19]. A better understanding of Wnt signaling activation is re-quired to develop a ‘tailored’ treatment strategy that specifically targets the relevant com-ponents.

  1. In discussion, since there are 19 members in Wnt family, how many potential non-canonical Wnt ligands may be possibly involved in the inhibitory effects of LGK974 anti-inflammatory response?

This is an interesting question. We have preliminary data that show that Wnt9, as a further non-canonical Wnt ligand, might also be released from cardiomyocyte under hypoxic conditions. However, measuring Wnt secretion, especially of very low expressed Wnt ligands, is quite challenging. The full ‘Wnt code’ of hypoxic/challenged cardiomyocytes still needs to be deciphered. We added the following paragraph to the discussion:

We could show that LGK974 reduces the secretion of the non-canonical Wnt ligand Wnt5a from cardiomyocytes, which could contribute to the anit-inflammatory effects. However, we cannot exclude that inhibiting the secretion of other non-canonical Wnt signaling as-sociated ligands may also be involved. The non-canonical Wnt9A and B, for instance, have been shown to be upregulated in a cardiac cryoinjury model in neonatal mice [32]. Wnt9A has been shown to correlate with pro-inflammatory cytokines [33]. The full Wnt ligand secretion pattern in cardiomyocytes still needs to be elucidated.

3.Canonical Wnt signaling pathway may need to be detected in the in vitro cell model, such as the change of beta-catenin level before and after LGK974 treatment. 

We agree that blockade of Wnt secretion by LGK-974 may also affect canonical Wnt signaling. In vitro, we do see a decline in active beta catenin levels (ABC) in cardiomyocytes when treated with LGK-974 compared to the DMSO control group(see below). However, ABC was not detectable in monocytes/macrophages in our setup.
